# Presence of Drug Interaction Between Penicillin and Hormonal Contraceptives in Women: A Scoping Review

**DOI:** 10.3390/healthcare13121364

**Published:** 2025-06-06

**Authors:** Jennifer Reis-Oliveira, Alex Junio S. Cruz, Nathalia S. Guimarães, Mauro Henrique N. G. Abreu

**Affiliations:** 1Department of Community and Preventive Dentistry, School of Dentistry, Universidade Federal de Minas Gerais, Belo Horizonte 31270-800, MG, Brazil; jenniferreisdeoliveira@hotmail.com (J.R.-O.); junio.alex@hotmail.com (A.J.S.C.); 2Department of Nutrition, School of Nursing, Universidade Federal de Minas Gerais, Belo Horizonte 30130-100, MG, Brazil; nasernizon@gmail.com

**Keywords:** penicillin, contraceptive agents, drug interactions, women’s health

## Abstract

Drug interactions (DIs) can alter the effects of medications or result in adverse reactions. This scoping review aimed to map the existing scientific evidence regarding the DI between penicillin and hormonal contraceptives in women. Following the PRISMA-ScR, we conducted electronic searches in the MEDLINE, Embase, Web of Science, and the Virtual Health Library databases from August 2023 to January 2024. Observational studies, monographs, dissertations, theses, and conference abstracts with adolescent, adult, and elderly women who were concurrently using hormonal contraceptives and penicillin were eligible. DIs were defined based on the signs/symptoms presented by the women and self-reported pregnancies. Of the 4023 records identified in the databases, nine studies, published between 1979 and 2021, were included. Variability was found among the studies in terms of sample size, data collection method, participant’s age, medication types, diagnostic approach used to confirm the pregnancy, and the author’s recommendation of the DI. It could therefore be concluded that while the literature suggests a DI between hormonal contraceptives and penicillin, the level of scientific evidence is still scarce. Additional research on systemic and population factors is essential to better understand this DI and its repercussions.

## 1. Introduction

Penicillin is an essential antibiotic that is effective in the treatment and prophylaxis of various bacterial diseases, thus improving patients’ quality of life [1,2]. Penicillin stands as a pivotal antibiotic, its discovery having profoundly revolutionized global health and consistently saving innumerable lives across decades [3,4]. Its notable efficacy in both the treatment and prophylaxis of diverse bacterial infections has significantly enhanced patient quality of life [5]. The increasing rate of antibiotic prescriptions in recent years has raised significant concerns regarding the emergence and spread of antimicrobial resistance (AMR). Klein et al. [6] reported a 39% rise in antibiotic consumption across 76 countries between 2000 and 2015. Additionally, following the Coronavirus Disease 2019 (COVID-19) pandemic, an increase in antibiotic prescriptions was observed in Brazil [7,8]. A survey conducted by the Federal Pharmacy Council of Brazil indicated that between 2019 and 2020, antibiotic consumption surged by over 60% in the Southeast, Northeast, and Central-West regions, while the North region experienced an alarming 123% rise [9]. Dos Santos et al. [10] further corroborated this trend, reporting a 21.3% increase in antibiotic consumption in 2020 compared to the previous year.

In contrast to this escalating consumption, the rational use of antibiotics remains essential for preserving their efficacy and mitigating the growing threat of AMR [11]. Antimicrobial Stewardship (AMS) programs, which focus on educating healthcare professionals about appropriate prescribing practices and increasing public awareness, have demonstrated effectiveness in curbing unnecessary antibiotic use, particularly in high-income countries [12]. Of particular note, the United Nations High-Level Meeting on Antimicrobial Resistance convened global leaders to establish international commitments and targets, including a 10% reduction in AMR-related mortality by 2030 and improved antibiotic stewardship in human healthcare [13]. Both interventional and observational studies indicate that AMS initiatives can significantly reduce antibiotic consumption, thereby alleviating the selective pressure that drives resistance. For instance, systematic reviews and meta-analyses suggest that AMS programs can lead to a 15% to 30% reduction in antibiotic use in hospital settings, a decline that is directly associated with lower rates of resistant bacterial strains [14]. AMR is not merely a distant future threat but an urgent global health crisis, with devastating consequences already evident in terms of increased mortality and depleted healthcare resources. Ensuring the rational use of antibiotics remains a cornerstone of global efforts to combat this crisis [11].

Despite their effectiveness, the indiscriminate use of antibiotics has reached large scales, resulting in treatment failures brought on by the emergence of multidrug-resistant organisms and thus to higher costs for new drugs [15,16,17,18]. Additional challenges include drug interactions, defined as pharmacological or clinical responses caused by the interaction of drug–drug, drug–food, drug–chemical substance, drug–medicinal plant, and drug–disease. These interactions could change the medication’s intended effects or result in adverse reactions [19,20]. While no other medications directly alter gut microbiota through antibiotic interaction, Nonsteroidal Anti-inflammatory Drugs (NSAIDs) and Proton Pump Inhibitors (PPIs) can exacerbate or modulate antibiotic-induced microbiota changes. Administered concurrently or sequentially, these drugs may impact microbial composition, potentially amplifying dysbiosis or affecting gut recovery [21,22].

In this scenario, one type of drug interaction that is under discussion and has produced some controversy in the literature is the concomitant use of penicillin with hormonal contraceptives (whether oral or injectable). Hormonal contraceptives consist of a combination of hormones, such as estrogen and progesterone, which, when introduced into the body, are metabolized in the liver. Estrogen is converted into estrogenic metabolites, some of which are hydrolyzed by enzymes in the intestinal bacteria, releasing active estrogen. The released estrogen is then reabsorbed into the bloodstream, increasing plasma estrogen levels and establishing the enterohepatic circulation [23]. When penicillin is ingested while using hormonal contraceptives, the antibiotic impacts the intestinal microbiota, potentially reducing the activity of enzymes necessary for the release of active estrogen into circulation, thereby decreasing hormone levels [24,25]. Despite the biological basis of this interaction, controversy persists regarding its impact on unintended pregnancies. Aronson and Ferner [26] stated that unintended pregnancies were seven times more frequent among women using antibiotics and hormonal contraceptives, making it necessary to warn them about the potential reduction in contraceptive effectiveness during antibiotic use. In their study, when analyzing reports of unintended pregnancies, they considered diarrhea as a possible confounding factor in interactions. However, although diarrhea caused by antibiotics might have reduced the effectiveness of the oral contraceptives, it did not show up as a signal. Therefore, it cannot be considered a confounding factor in this data. By contrast, a study carried out based on the United States Preventive Services Task Force concluded that there is no decrease in the effect of hormonal contraceptives when used simultaneously with penicillin [27]. These two studies [26,27] presented conflicting findings. It is crucial to highlight that this divergence likely stems from fundamental differences in their methodologies. The investigation by Aronson and Ferner [26] was an analysis of reported suspected adverse drug reactions, whereas Simmons et al. [27] conducted a systematic review concentrating on primary studies. Such variations in study design and analytical approaches can account for the observed variability in results. Nevertheless, despite the potential implications of this interaction, few studies have investigated its frequency in the general population. Moreover, the growing use of hormonal contraceptives reflects women’s concerns about unintended pregnancies [28].

Given this context, the present study aimed to map the existing scientific evidence regarding the drug interaction (DI) between penicillin and hormonal contraceptives in women. Additionally, considering the growing global public health crisis due to antibiotic resistance and polypharmacy, expanding knowledge about the characteristics associated with the prescription of these medications may support the planning of interventions to address the issue [29,30,31]. While this review primarily examines the drug interaction between penicillin and hormonal contraceptives, it is important to acknowledge that certain included studies also explore other subclass antibiotics within the beta-lactam group. Throughout the manuscript, we will explicitly differentiate references to penicillin itself, other beta-lactams, and antibiotics from distinct classes.

## 2. Materials and Methods

This scoping review was developed in accordance with the Joanna Briggs Institute (JBI) tool and reported according to Preferred Reporting Items for Systematic reviews and Meta-Analyses extension for Scoping Reviews (PRISMA-ScR) (Appendix A) [32]. This review was previously registered on the Open Science Framework platform (OSF)—<https://osf.io/rux2v/?view_only=6721c594caae4be99be2f84bbead9e31 (accessed on 10 January 2024)>.

The scoping review aimed to map all the scientific evidence on identifying DIs between penicillin and hormonal contraceptives in women. To construct the guiding research question, we used the PCC acronym, where P (participants) = adolescent, adult, and elderly women; C (concept) = drug interactions (DIs); and C (context) = the use of hormonal contraceptives and penicillin.

### 2.1. Exposure

The use of penicillin was considered a form of exposure. Penicillin belongs to the beta-lactam subclass of antibiotics, which includes, but is not limited to, amoxicillin, ampicillin, carbenicillin, dicloxacillin, nafcillin, oxacillin, penicillin G, penicillin V, piperacillin, and ticarcillin. Studies examining the interaction of hormonal contraceptives with other beta-lactam antibiotics beyond penicillin were also considered, provided their findings were relevant to the broader discussion of interactions within this subclass [33].

### 2.2. Outcome

The outcome was the interaction between concurrent use of hormonal contraceptives and penicillin. DIs will be defined as signs and symptoms presented by women and self-reported pregnancies.

### 2.3. Eligibility Criteria

The inclusion criteria for this study were adolescent, adult, and elderly women that used hormonal contraceptives (oral or injectable) and penicillin concurrently; observational studies; monographs, dissertations, theses, and conference abstracts. The exclusion criteria were infants, children (under 13 years of age), pregnant women, and men. Studies without restrictions on language or publication date were included. Letters/editorials or protocols without results, ecological studies, and literature reviews were excluded.

### 2.4. Information Sources and Search Strategies

The search strategy was developed using keywords indexed in the Medical Subject Headings (MeSH) descriptor database. The search syntaxes were adapted according to Emtree and DeCS descriptor bases, and the databases searched were as follows: PubMed/Medline, Embase, Virtual Health Library (LILACS, ADOLEC, BBO, BDENF, HISA, LEYES, MEDCARIB, REPIDISCA, OPAS, WHOLIS, and DESASTRES), and Web of Science (S1). 

### 2.5. Study Selection

Once identified, articles were imported into the reference management tool Rayyan (Qatar Computing Research Institute, Doha, Qatar), and duplicate studies were removed. Two reviewers independently selected studies by title and abstract for inclusion or exclusion. After this phase, all included articles were evaluated by reading the full text, considering the eligibility criteria. Differences were discussed until a consensus was reached. In cases where there was no consensus regarding inclusion or exclusion, the decision was made by a third review author.

### 2.6. Data Extraction

Two independent review authors conducted data extraction, and disagreements were discussed until a consensus was reached. The extracted data includes study characteristics (reference and journal), population characteristics (context, sample, sampling, name, and dosage of antibiotics), and main results (frequency of use, dose, and duration of prescribed penicillin). Signs and symptoms presented by women and self-reports about pregnancies were defined as DIs. A standardized and piloted form, developed in Excel (Microsoft, Office 365), was deployed for data extraction.

### 2.7. Data Synthesis

Data synthesis consisted of two stages. The PRISMA-ScR flowchart was used for the first stage of data synthesis to achieve the results of the study eligibility process.

The second stage involved filling out the extraction table in a narrative form with all the data found in the research. Subgroup analysis was performed considering the type of study design, age range of the women evaluated, and the medications used, together with their prescribed dose and duration. No specific scales were used to assess the methodological quality of the included articles due to the nature of the topic investigated, that is case report studies or case series, which also makes it difficult to select an appropriate assessment instrument. Possible methodological flaws in the studies were analyzed based on measurement bias, selection bias, and confounding bias [34].

## 3. Results

Searches in electronic databases retrieved 4023 references. After the removal of duplicates (n = 179), 3844 titles and abstracts were evaluated. The full texts of 11 references were then screened, but one reference could not be retrieved. Ten references met the eligibility criteria and eight were included, and the two excluded studies either had no data available for data extraction (n = 1) or did not assess interest variables (n = 1). One reference was identified manually through a citation search and was included. A flowchart of the selection process is shown in Figure 1.

### 3.1. Characteristics of Included Studies

Of all the included studies, three were case series and case reports [35,36,37], three were cross-sectional [26,38,39], two were case-crossover [40,41], and one was drug adverse surveillance analysis [42]. The included articles were from 1979 to 2021. The predominant language was English (n = 8), but there were also articles in Italian (n = 1). No study with a cohort design was identified. The studies were conducted in Italy, the United States of America (USA), Denmark, and the United Kingdom (UK) (Table 1 and Table 2).

### 3.2. Results of Individual Studies

All the studies analyzed potential DIs with oral hormonal contraceptives [26,35,36,37,38,39,40,41,42]. Among those, five mentioned the existence of an interaction between penicillin and hormonal contraceptives [15,21,22,23,25]. Back et al. [42] registered 32 self-reports of DIs between penicillin and oral contraceptives (OC) from 63 registers of pregnancies in women who received antibiotics, but they made no recommendations in their study.

Some of the studies that were evaluated in this review did not provide data of interest, e.g., three did not specify the age of the participants [26,42], one did not mention the source of information [36], and two did not confirm any unintended pregnancy [39,40].

Of the nine studies selected, five concluded and indicated that women should use another method of contraception when using penicillin and hormonal contraceptives concurrently [26,35,37,39,40]. Bainton [35] posited a causal link between contraceptive failure in a reported case and the established mechanism by which penicillin perturbs the gut microbiota, potentially leading to a reduction in circulating estrogen levels, which could subsequently result in escape ovulation. In Bainton’s investigation, this association was inferred from a clinical case presentation and an understanding of the proposed mechanism involving the enterohepatic circulation of estrogens. Direct measurement data of hormonal levels were not presented in this specific case report. The intricate interplay between estrogen and androgenic hormones, mediated by enzymes such as aromatase, plays a pivotal role in the regulation of circulating hormone levels. This process significantly influences crucial physiological functions, including sexual development, bone health, and cardiovascular integrity. Dysregulation in this conversion can lead to hormonal imbalances, contributing to conditions such as insulin resistance and osteoporosis. Furthermore, various factors, including pharmacotherapy, endocrine disorders, and alterations in the gut microbiota, can interfere with this enzymatic mechanism, thereby modifying systemic hormone concentrations [24]. While the broader discussion concerning the conversion between estrogens and androgenic hormones was not the primary focus of the studies included in this review, we acknowledge the inherent complexity of hormonal metabolism within this context.

The types of penicillin were more varied, whether administered orally or injected, with the most commonly cited being ampicillin [26,36,41]. In analytical studies [26,38], there were 5 to 62 cases of pregnancy in the DI group, as compared to 7 and 9 in the group with no DI.

The analyzed studies presented other medications that may have potential interactions with hormonal contraceptives, but with limited evidence, such as: nitrofurantoin, nevirapine, rifabutin, rifampicin, ritonavir, citalopram, topiramate, lansoprazole, loperamide, loratadine, propranolol, theophylline, zolpidem, tetracycline, phenytoin, phenobarbitone, primidone, carbamazepine, ethosuximide, sodium valproate, cotrimoxazole, metronidazole, cephalosporins, trimethoprim, erythromycin, sulphonamides, griseofulvin, fucidic acid, co-trimoxazole, isoniazid, amphotericin, oxazepam, azatadine, cortisone, indomethacin, ibuprofen, paracetamol, aspirin, prochlorperazine [26,36,39,41,42].

## 4. Discussion

This review identified only a limited number of studies that provided data on DIs between penicillin and hormonal contraceptives. Among these manuscripts, stronger scientific evidence was found in the analytical case-crossover studies and the cross-sectional studies. No cohort studies were identified. Based on this synthesis, it is necessary to point out that the level of evidence is quite low [43].

Most of the primary studies included in this review support the existence of a DI between penicillin and hormonal contraceptives, with several of them noting unintended pregnancies as a consequence [26,35,36,37,38,39,42]. These studies are case reports and case series, cross-sectional studies, and drug adverse surveillance analyses, published between 1979 and 2021. Only the articles by Helms et al. [38] and Aronson and Ferner [26] included a control group in their studies. Aronson and Ferner [26] explained that ampicillin reduced circulating estrogens, by interfering with enterohepatic circulation, and that they also increased fecal concentrations of conjugated endogenous estrogens in pregnant women up to 70 times in 2–3 days. The same findings were reported by Turcato and Correa [44]. Another factor that may explain DI is that OCs with ethyl estradiol lower than 35 µg (OC with lower doses) contributed to a higher incidence of failure rates, leading to unintended pregnancies. For Trussell and Portman [45], this hypothesis would not be the most plausible. In their study, they conducted a thorough survey of the Pubmed and Management Information System databases and found that higher doses of OCs were the most commonly prescribed during the period when declines in contraceptive efficacy were noted.

Studies with higher levels of evidence typically use more rigorous methodologies to mitigate bias. However, even high-quality studies can present bias, and the quantitative analysis of bias is an important tool for identifying and correcting these distortions [34]. When considering measurement bias, studies that portray case reports and series, as well as those using retrospective data, are more susceptible to this type of bias Back et al. [42] evaluated ‘yellow card’ reports—spontaneous notifications of suspected adverse drug reactions submitted by healthcare professionals in the United Kingdom to the Committee on Safety of Medicines between 1968 and 1984. This study may be subject to measurement bias as these reports are subjective and depend on the perception of the reporting professional. Another study that may also have measurement bias, for the same reason—uncertainty whether data collection was performed consistently or with underreporting of cases—is that of Aronson and Ferner [26]. The studies by Russo et al. [36], Silber [37], and Bainton [35], which are case reports and case series, depict characteristics described by patients and pregnancy itself, which, not being directly controlled by the researchers, can suffer from measurement bias.

Two studies (22.2%) found no DIs between penicillin and hormonal contraceptives and that, according to self-reports, women could not have become pregnant through DIs [40,41]. These studies were case-crossovers, published between 2011 and 2018, in which the authors state that this type of interaction cannot lead to unintended pregnancies. Toh et al. [41] also corroborate the same thinking when they state that, after evaluating data from the Slone Epidemiology Center Birth Defects Study (BDS) and the National Birth Defects Prevention Study (NBDPS) databases, no association was found between antibiotic use and the risk of pregnancy among OC users. In the literature, we found studies mentioning that only the antibiotic rifampin could decrease the effectiveness of hormonal contraceptives [46,47].

The interaction between penicillin and hormonal contraceptives is frequently attributed to modifications in gut microbiota, which may diminish the enterohepatic circulation of estrogens [44]. However, existing literature also explores additional mechanisms of antibiotic–drug interactions, such as the induction or inhibition of hepatic enzymes responsible for metabolizing contraceptive hormones. For instance, rifampicin, a non-penicillin antibiotic, is well-documented for its ability to reduce the efficacy of hormonal contraceptives through hepatic enzyme induction [48]. Although this study primarily examines the effects of penicillin, the broader scope of antibiotic–contraceptive interactions encompasses multiple mechanisms. Pharmacokinetic research investigates direct interactions between drugs and metabolites, aiming to elucidate complex metabolic pathways [49]. A comprehensive understanding of these additional mechanisms is essential for refining clinical recommendations and enhancing patient safety. Although pharmacokinetic studies are not the central focus of this scoping review, they remain critical for clarifying the precise pathways by which drugs interact within the human body. Integrating pharmacokinetic findings with clinical outcome data is essential for a more accurate assessment of the risk of drug interactions [50,51].

Pottegard et al. [40] used Danish national records from the Abortion Registry and Medical Birth Registry between the years 1997 and 2015. When using secondary data in pharmacoepidemiologic studies, attention must be paid to the issue of biases within the studies as they can interfere in the conclusions. In the study in question, the authors themselves mention that time-related confounding factors could be a limitation, such as including changes in sexual activity in relation to infection or the temporary use of alternative contraceptive methods due to a fear of interaction with dicloxacillin, which, in this case, would interfere with the study estimates. One confounding factor that could interfere with the estimates of a study and was not mentioned is the healthy user/adherer effect, for example. It is believed that access to healthcare resources is associated with a higher level of education and health-seeking behavior [52]. It may well be that more well-educated women use the medication more correctly, reducing the chances of possible failure.

While there are disagreements about the existence of DIs between penicillin and hormonal contraceptives, majority of the papers recommended using supplementary contraceptive methods together with penicillin [26,35,37,39,40,41]. Studies highlighted that, due to DI, it is necessary to use additional barriers as the effectiveness of hormonal contraceptives decreases during the period of use of this class of antibiotics [26,35,37,39]. Despite affirming that there is no interaction, Toh et al. [46] and Pottegard et al. [40] recommend the use of complementary physical barrier methods after discontinuing penicillin, at least until further studies confirm the lack of association. These recommendations highlight the importance of considering the use of barrier contraceptive methods, such as condoms, during antibiotic use so as to prevent contraceptive failure and unplanned pregnancies.

When it comes to analyzing the DIs of penicillin and hormonal contraceptives, one must remember that it is necessary to consider the exposure window, defined as the 15 days before and 15 days after the probable date of conception. Within this period, it is important to analyze the medications the woman has used [53]—in this review, penicillin. Notably, only the studies by Helms et al. [38], Toh et al. [41] and Pottegard et al. [40] adequately considered this factor. Although it is extremely important to study DIs and their potential consequences, it is difficult to assess DIs, mainly due to the number of drugs used by a single patient [54], and it is often impossible to isolate the adverse reactions of the drugs of interest.

The existing data on drug interactions between penicillin and hormonal contraceptives predominantly stem from reports and case series. However, these studies often include a very limited number of participants and lack appropriate control groups. Within the pyramid of evidence, reports and case series are one of the lowest levels of evidence [29]. Given these limitations, there is a compelling need for more rigorous studies that provide higher levels of evidence, enabling a more precise assessment of the DI. Additionally, studies employing animal models also contribute significantly to understanding the underlying mechanisms of drug interactions, offering valuable insights into metabolic pathways and the physiological intricacies of the interaction. Nevertheless, these studies were not eligible for direct inclusion in this review due to our exclusive focus on human populations [33]. It is crucial to acknowledge that the absence of evidence found in several studies does not necessarily prove an absence of association. This underscores the need for further research, since unintended pregnancies can significantly change people’s lives.

Official recommendations for the co-prescription of antibiotics and hormonal contraceptives vary among different health agencies and clinical guidelines. Generally, guidelines recognize that only rifampicin and rifabutin, which are enzyme-inducing antibiotics, have a proven significant impact on the effectiveness of hormonal contraceptives, thus requiring specific counseling or an alternative contraceptive method [55]. For other classes of antibiotics, including penicillin, many health agencies, such as the U.S. Centers for Disease Control and Prevention (CDC) and the UK’s Faculty of Sexual and Reproductive Healthcare (FSRH), state that there is insufficient evidence to routinely recommend the use of additional contraceptive methods, although cautious advice to use a barrier method for patient reassurance may be considered [56,57]. Our findings corroborate the inconsistency in the literature regarding penicillin, indicating the need for clearer and more robust evidence-based guidelines.

## 5. Conclusions

It can thus be concluded that, although the literature suggests a potential drug interaction between hormonal contraceptives and penicillin, the level of scientific evidence is limited and often methodologically weak, with a predominance of case reports and case series. The inconsistency of findings and the prevalence of studies with potential methodological biases restrict the ability to draw definitive conclusions regarding the frequency and clinical impact of this interaction at the population level. In light of these considerations, further research—incorporating systemic and population-level factors and employing more robust study designs (e.g., cohort studies or controlled clinical trials, where ethically feasible)—is both justified and essential to better understand this type of drug interaction and its potential implications, particularly concerning unintended pregnancies, which can have significant consequences in women’s lives.

## Figures and Tables

**Figure 1 healthcare-13-01364-f001:**
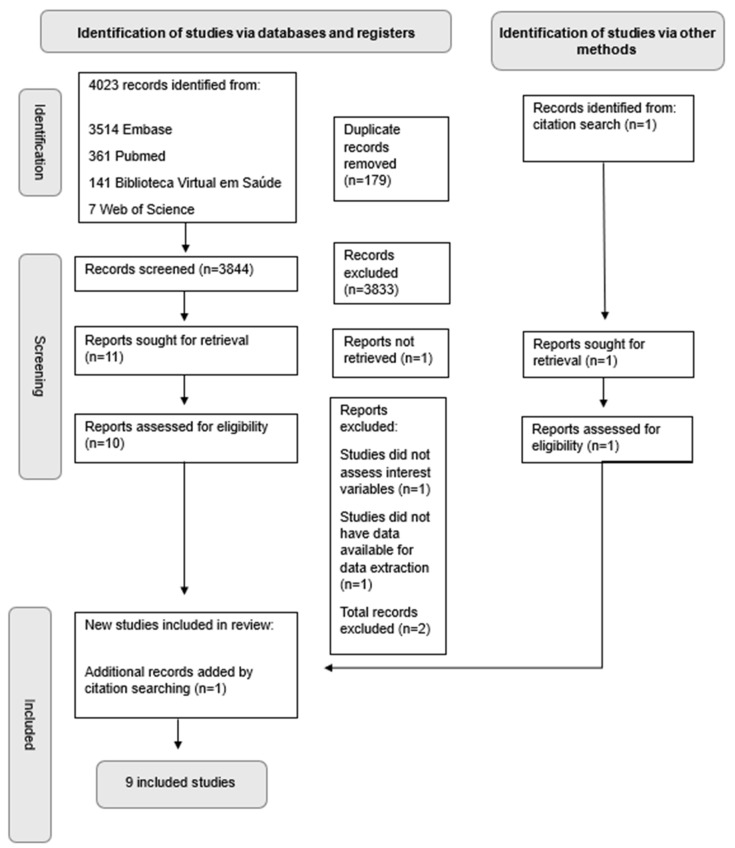
Flowchart for the systematic article selection process.

**Table 1 healthcare-13-01364-t001:** Characteristics of included studies in the scoping review.

	Authors	Year	Journal	Study Design	Objectives of the Study	Duration of Study	Sample Size	Age (Years)	Country	Source of Information	Unintended Pregnancy Confirmed	Exposure	Hormones Present in Contraceptives
1	Russo S, Tapparelli E, D’alessio L. [36]	1979	Clin ExpObst Gyn	Case series	Interactions between steroidal oral contraceptives and other drugs, with a special focus on antibiotics	No information	4 patients with DI	Mean = 24.8	Italy	No information	Pregnancy test	OC * and oral ampicillin	Norgestrel and Ethinylestradiol
2	Silber, TJ. [37]	1983	J Adolesc Healthcare	Case report	To report a clinical case of possible contraceptive failure associated with the concomitant use of an antibiotic (oxacillin) and an oral contraceptive in an adolescent	Not applicable	1 patient	18	USA	Adolescent clinic	Beta subunit human chorionic gonadotropin in serum test	OC and oxacillin sodium	Norethindrone and estradiol
3	Bainton, R. [35]	1986	Oral Surg Oral Med Oral Pathol	Case report	To report a clinical case of contraceptive failure associated with the concomitant administration of a long-acting penicillin and a low-dose oral contraceptive, and to discuss the possible pharmacological mechanisms of this interaction	Not applicable	1 patient	19	UK	Oral surgery clinic	Pregnancy test	OC and fortified benethamine penicillin injection	Ethinyl estradiol and levonorgestrel
4	Back et al. [42]	1988	Br. J. Clin. Pharmac.	Drug adverse surveillance analysis.	Analysis of adverse reactions to evaluate data on unintended pregnancies in women using OC concomitantly with antiepileptic drugs or antibiotics	1968–1984	All yellow cards (with drug adverse reaction)	No information	UK	All yellow cards which indicated a pregnancy in a woman taking an OC plus an antibiotic	Self-report	OC and penicillin	Progestogen, progestogen with estrogens, biphasic, and triphasic preparations—reports indicate that some contained ethinylestradiol and levonorgestrel, though the specific combinations were not specified.
5	Kovacs et al. [39]	1989	Med J Aust	Cross-sectional	To determine the factors associated with unintended pregnancies in OC users and to compare the percentage of OC types used by patients with unintended pregnancies with the national market usage distribution during the same period	1985–1986	72 patients with DI	Range-18–38 years	Australia	The Family Planning Association’s clinics and several centers which undertook terminations of pregnancy	No report	OC and penicillin	Triphasic, 30-µg ethinyloestradiol, 50-µg ethinyloestradiol, norethisterone, and progesterone
6	Helms et al. [38]	1997	J Am Acad Dermatol.	Cross- sectional	To examine the effect of commonly prescribed oral antibiotics (tetracyclines, penicillins, cephalosporins) on the failure rate of OC	No information	356 patients	15–19 years of age = 77; 20–24 years of age = 118; 25–29 years of age = 77; 30–34 years of age = 64; 35 and older = 20	USA	Review of records in three private dermatology offices	Chart record (if reported by the patient to the physician during regular visits) as well as by the survey responses, and confirmed with a follow-up telephone call	OC and oral antibiotics (minocycline, cephalosporin, and penicillin)	No information
7	Toh et al. [41]	2011	Contraception	Case- crossover	To examine the effect of concomitant antibiotic treatment on the risk of breakthrough pregnancy among users of combined OC	1997–2008	1330 patients	<20 = 225; 20–24 = 440; 25–29 = 353; 30–34 = 215; ≥35 = 97; Unknown = 0	USA	Participants of the Slone Epidemiology Center Birth Defects Study (BDS) and the National Birth Defects Prevention Study (NBDPS)	Trained interviewers using a computer-assisted telephone interview	OC and oral antibiotics	It mentions that they are combined oral contraceptives and that it excluded users of progestin-only OC
8	Pottegård et al. [40]	2018	Basic Clin Pharmacol Toxicol	Case-crossover	To investigate whether the use of dicloxacillin is associated with an increased risk of unintended pregnancy among users of OC	1997–2015	364 women	Median 23 [interquartile range (IQR) 19–29]	Denmark	The researchers sampled women with seemingly unintended pregnancy from two separate data sources: the Abortion Registry (elective abortion) and the Medical Birth Registry (women giving birth)—1997–2015. For both groups we found that they had used oral contraceptives at the time of conception.	No information	OC and dicloxacillin	No information
9	Aronson JK, Ferner RE. [26]	2021	BMJ Evid Based Med	Cross-sectional	To reassess the hypothesis that non-enzyme-inducing antibiotics (such as penicillins, cephalosporins, tetracyclines, etc.) reduce the effectiveness of OC, leading to unintended pregnancies	Not applicable	74,623 with DI	No information	UK	Reports of suspected adverse drug reactions published by the Medicines and Healthcare products Regulatory Agency (MHRA) for information on suspected adverse drug reactions between 1963 and July 2018. To be included, reports had to mention exposure to a drug of interest linked to a report of an unintended pregnancy	Self-report	OC and oral antibiotics (amoxicillin, ampicillin, cephalexin, ciprofloxacin, erythromycin, metronidazole, nitrofurantoin, oxytetracycline, and trimethoprim)	No information

* OC: oral contraceptive.

**Table 2 healthcare-13-01364-t002:** DI and unintended pregnancy.

	Name of the Medicine	Dose and Duration of Prescribed Penicillin	Number of Women Who Became Pregnant	Author’s Recommendations
1	All patients (n = 4) received ampicillin and information on OC * was available for only one woman (Norgestrel 0.25 mg + Etinilestradiol 0.05 mg)	No information	1 pregnancy from 4 patients	Need to carefully avoid the simultaneous prescription of antibiotics and hormonal contraceptives and inform the patient of this possibility. It can be added that not only the listed drugs, but also the use of any other drug, while taking oral contraceptives, must be carefully screened. It is believed that new studies can highlight new interference
2	OC: mg norethindrone/0.035 mg estradiol. Antibiotics: oxacillin sodium	500 mg oxacillin sodium every 6 h for 6 weeks	1 pregnancy	Advising a different hormonal contraceptive method or an additional contraceptive modality may be indicated in adolescents taking long-term antibiotic medication
3	OC: ethinyl estradiol 30 mcg and levonorgestrel 150 mcg; Antibiotic: fortified benethamine penicillin injection	Benethamine penicillin 500,000 I.U.; Procaine penicillin 250,000 I.U.; Benzyl penicillin 500,000 I.U	1 pregnancy	Advising a different hormonal contraceptive method or an additional contraceptive modality may be indicated in adolescents taking long-term antibiotic medication
4	OC: estrogen, progestogen, biphasic, triphasic, pill. Antibiotics: penicillin	No information	32 self-reports of DI between penicillin and OC from 63 registers	No recommendation
5	OC: triphasic, ethinyloestradiol 30 mcg, ethinyloestradiol 50 mcg, norethisterone, progesterone-only. Antibiotics: penicillin	No information	22 self-reports of DI between penicillin and OC from 72 registers	Additional contraceptive precautions must be taken until at least seven continuous tablets have been taken after an episode which may impair the efficacy of the drug
6	OC: not reported; antibiotics: minocycline (tetracyclines)—3 patients, cephalosporin (cephalosporins)—2 patients and no penicillin	No information	5 self-reports of DI between penicillin and OC from 356 registers	The difference in failure rates of OCs when taken concurrently with antibiotics commonly used in dermatology versus OC use alone suggests that these antibiotics do not increase the risk of pregnancy. Physicians and patients need to recognize that the expected OC failure rate, regardless of antibiotic use, is at least 1% per year; it is not yet possible to predict in whom OCs may fail
7	OC: not reported; antibiotics: ampicillin/amoxicillin and others	No information	There is no difference in the frequency of use of ampicillin/amoxicillin between the case period (4 weeks before conception) and the control period (4–8 weeks before conception)	No association was found between concomitant antibiotic use and the risk of breakthrough pregnancy among OC users. However, due to limited power and potential carryover effects, findings from the study cannot rule out an elevated risk of OC failure among antibiotic users
8	OC: combination oral contraceptives, progestogen oral contraceptives; antibiotics: dicloxacillin	No information	There is no difference in the use of dicloxacillin at the time of ovulation and control windows	The results implied no association between the use of dicloxacillin and the risk of oral contraceptive failure. However, oral contraceptive failure may have severe social consequences. It is suggested supplementary physical barrier methods be used until 2 weeks after discontinuation of dicloxacillin, at least until further studies confirm the lack of any association
9	Oral contraceptive and oral antibiotics (amoxicillin, ampicillin, cephalexin, ciprofloxacin, erythromycin, metronidazole, nitrofurantoin, oxytetracycline, and trimethoprim)	No information	46 unintended pregnancies (62 per 100,000, (95% CI 44 to 79) in those who took antibiotics (amoxicillin, ampicillin, cephalexin, ciprofloxacin, erythromycin, metronidazole, nitrofurantoin, oxytetracycline, and trimethoprim). Antibiotics increased the odds of unintended pregnancy by 6.7-fold	This study provides a signal that antibacterial drugs may reduce the efficacy of hormonal contraceptives. Women taking hormonal contraceptives should be warned that antibiotics may impair their effectiveness. Extra precautions can be taken during a course of antibiotics; an unintended pregnancy is a life-changing event

* OC: oral contraceptive.

## Data Availability

All data generated or analyzed during this study are included in this published article.

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
