# Peer review of "Presence of Drug Interaction Between Penicillin and Hormonal Contraceptives in Women: A Scoping Review"

_healthcare, 2025, doi:10.3390/healthcare13121364_

Round 1
Reviewer 1 Report
Comments and Suggestions for Authors
The authors have done a great job at identifying an important area of research that is significantly lacking in data: whether antibiotics impact hormonal contraceptives. Overall, the paper reads well and is very interesting. My comments for improvement mainly focus on use of language and clarity.
Additionally, I’d like to highlight that the authors have primarily focused on penicillin but in some cases have included other antibiotics of the same subclass, beta-lactams. I would recommend the authors address this early in the article, and ensure that it is clear each time whether penicillin, or another beta-lactam antibiotic, or a non-beta-lactam antibiotic, is being referred to. Furthermore, it would be useful for the authors to mention whether other antibiotic classes have been found to interact with oral contraceptives or other drugs, through the mechanism of reducing the microbiota’s capacity for drug metabolism. It would also be interesting to hear the authors’ opinion on whether other mechanisms of drug interaction may be possible other than solely through the microbiota (e.g., direct drug-drug interactions) – has this been investigated before? A sentence or two in the discussion addressing this would be great.
Line 29 – the authors have here understated the importance of penicillin – I would suggest to address how the discovery of penicillin changed the landscape of healthcare globally and has since saved lives.
Line 29-30, and throughout this paragraph – the increased rate of antibiotic prescriptions is concerning because of increasing antibiotic resistance – I suggest the authors include this sentiment and additionally provide a statistic on how antibiotic stewardship is essential to maintain our current standard of healthcare, rather than focusing too much on the increase of antibiotic prescription. Antibiotic resistance is one of the major hurdles facing healthcare in future years, thus I recommend the authors to focus more strongly on this throughout the paper.
Lines 32-35: “some growth” is very arbitrary. I suggest the two sentences to be merged and the authors to focus specifically on examples like the increase in Brazil.
Lines 37-41: The authors have not yet spoken about whether antibiotics are known to have other drug interactions or if there is existing evidence suggesting that there may be. I suggest the authors to include a sentence or two about any existing known drug interactions with antibiotics, (particularly through the mechanism of reducing the resident microbiota) or if there is none known, that the authors identify this and highlight the need for more research.
Line 44 – “whether oral or injectable” – it is not clear whether the authors are referring to antibiotics or hormonal contraceptives here
Line 44 – not all hormonal contraceptives are a combination of hormones – for example progesterone-only contraceptives. Please ensure language is clear and accurate throughout.
Line 54 – did Aronson and Ferner also correct for/ test for confounding effects from the infection itself? Please mention briefly
Line 57 – are there any reasons why these two studies may have found conflicting results, such as study design? For example, a sentence like “these two studies found different results although it must be noted that study #1 included only X and not Y or Z”
Line 65: The sentence “The study’s findings intend to shed even greater light…” is redundant, authors can omit.
Line 66: “The public health crisis” that is referred to – please include references and/or more specific language to back up this statement, as it is very important to the study.
Lines 80-84: The authors have here identified other antibiotics in the subclass of beta-lactams, but have not specifically said whether or not these are included in the study – please revise language for clarity
Line 137 – I commend the authors use of this flowchart, it is very clear and easy to understand.
Lines 146 onward – this may be outside of the authors control, but this table needs to be re-formatted as it is extremely difficult to read in its current state (perhaps needs to be made landscape?). Also, first row – “ampiciline” please use English term. I also recommend that a column is included to detail what hormones were included in each contraceptive – eg. Progesterone only, combined pill, etc. This has been included but only for some studies, so please make it consistent throughout the table. May be easier to include a separate column as I have mentioned.
Line 148 onward – Row 5 of Table 2 – “the Pill” – this term is not used elsewhere in the article, so please ensure consistent language throughout. Also, the term “OC” is used in this table but not in table 1, please ensure consistency throughout the article for clarity.
Line 153 – It is not clear what is meant by “63 registers”, please clarify.
Line 155 – The sentence “Some data were not possible to be collected in the studies” suggests that the current authors may have had control over what data were collected in the studies. For clarity, I suggest the authors change the wording to something along the lines of “Some of the studies that were evaluated in this review did not provide data of interest, such as…”
Line 161 – Did Bainton show with evidence that circulating estrogen decreases with penicillin? Please clarify in the text, and include how this was done (number of patients, method of measuring hormone levels etc) as this would be important. Did overall circulating sex hormone decrease, or was it just estrogen (note also: the authors also have not addressed the interconversion of estrogen to androgen hormones, which may play into levels of circulating hormone)? Again, please clarify in the text. Furthermore, I suggest the authors avoid use of his/her pronouns and use “their” (i.e. “Bainton [22] associated contraceptive failure in their case report…)
Discussion – I commend that the authors have used this study to highlight a lack of focus into this area, and have consistently shown that although the existing evidence is limited, there must be more work done.
Lines 198 onward – please clarify what “yellow card reports” are
Comments on the Quality of English Languagesee above
Author Response
1) The authors have done a great job at identifying an important area of research that is significantly lacking in data: whether antibiotics impact hormonal contraceptives. Overall, the paper reads well and is very interesting. My comments for improvement mainly focus on use of language and clarity.
Authors’ response: Thank you so much for your positive feedback on the topic's relevance and the overall quality of our manuscript. We truly appreciate your suggestions regarding language use and clarity. We've thoroughly revised the manuscript to enhance the writing, standardize technical terms, and ensure the text is clearer and more precise, as you recommended.
2) Additionally, I’d like to highlight that the authors have primarily focused on penicillin but in some cases have included other antibiotics of the same subclass, beta-lactams. I would recommend the authors address this early in the article, and ensure that it is clear each time whether penicillin, or another beta-lactam antibiotic, or a non-beta-lactam antibiotic, is being referred to. Furthermore, it would be useful for the authors to mention whether other antibiotic classes have been found to interact with oral contraceptives or other drugs, through the mechanism of reducing the microbiota’s capacity for drug metabolism. It would also be interesting to hear the authors’ opinion on whether other mechanisms of drug interaction may be possible other than solely through the microbiota (e.g., direct drug-drug interactions) – has this been investigated before? A sentence or two in the discussion addressing this would be great.
Authors’ response: Thank you for your valuable feedback. We fully acknowledge the need for greater clarity and have carefully revised the text to explicitly distinguish references to penicillin, other beta-lactam antibiotics, and other antibiotic classes. Additionally, as per your suggestion, we have incorporated a discussion on alternative mechanisms of drug interaction beyond the gut microbiota.
Introduction section: “While this review primarily examines the drug interaction between penicillin and hormonal contraceptives, it is important to acknowledge that certain included studies also explore other subclass antibiotics within the beta-lactam group. Throughout the manuscript, we will explicitly differentiate references to penicillin itself, other beta-lactams, and antibiotics from distinct classes.”
Discussion section: “The interaction between penicillin and hormonal contraceptives is frequently attributed to modifications in gut microbiota, which may diminish the enterohepatic circulation of estrogens [46]. However, existing literature also explores additional mechanisms of antibiotic-drug interactions, such as the induction or inhibition of hepatic enzymes responsible for metabolizing contraceptive hormones. For instance, rifampicin, a non-penicillin antibiotic, is well-documented for its ability to reduce the efficacy of hormonal contraceptives through hepatic enzyme induction [50]. Although this study primarily examines the effects of penicillin, the broader scope of antibiotic-contraceptive interactions encompasses multiple mechanisms. Pharmacokinetic research investigates direct interactions between drugs and metabolites, aiming to elucidate complex metabolic pathways [51]. A comprehensive understanding of these additional mechanisms is essential for refining clinical recommendations and enhancing patient safety. Although pharmacokinetic studies are not the central focus of this scoping review, they remain critical for clarifying the precise pathways by which drugs interact within the human body. Integrating pharmacokinetic findings with clinical outcome data is essential for a more accurate assessment of the risk of drug interactions [52,53].”
1) Line 29 – the authors have here understated the importance of penicillin – I would suggest to address how the discovery of penicillin changed the landscape of healthcare globally and has since saved lives.
Authors´response: We agree that the significance of penicillin has been underestimated. We have revised the text to emphasize the transformative impact of its discovery on global health.
Introduction section: “Penicillin stands as a pivotal antibiotic, its discovery having profoundly revolutionized global health and consistently saving innumerable lives across decades [3,4]. Its notable efficacy in both the treatment and prophylaxis of diverse bacterial infections has significantly enhanced patient quality of life [5].”
2) Line 29-30, and throughout this paragraph – the increased rate of antibiotic prescriptions is concerning because of increasing antibiotic resistance – I suggest the authors include this sentiment and additionally provide a statistic on how antibiotic stewardship is essential to maintain our current standard of healthcare, rather than focusing too much on the increase of antibiotic prescription. Antibiotic resistance is one of the major hurdles facing healthcare in future years, thus I recommend the authors to focus more strongly on this throughout the paper.
Authors´response: We've taken your suggestion to heart and placed a stronger emphasis on the concern regarding antibiotic resistance and the critical importance of rational use. The text has been revised to incorporate this crucial perspective, offering a more robust context for the antimicrobial resistance crisis.
Introduction section: “The increasing rate of antibiotic prescriptions in recent years has raised significant concerns regarding the emergence and spread of antimicrobial resistance (AMR). Klein et al. [6] reported a 39% rise in antibiotic consumption across 76 countries between 2000 and 2015. Additionally, following the Coronavirus Disease 2019 (COVID-19) pandemic, an increase in antibiotic prescriptions was observed in Brazil. A survey conducted by the Federal Pharmacy Council of Brazil indicated that between 2019 and 2020, antibiotic consumption surged by over 60% in the Southeast, Northeast, and Central-West regions, while the North region experienced an alarming 123% rise [9]. Dos Santos et al. [10] further corroborated this trend, reporting a 21.3% increase in antibiotic consumption in 2020 compared to the previous year.
In contrast to this escalating consumption, the rational use of antibiotics remains essential for preserving their efficacy and mitigating the growing threat of AMR [11]. Antimicrobial Stewardship (AMS) programs, which focus on educating healthcare professionals about appropriate prescribing practices and increasing public awareness, have demonstrated effectiveness in curbing unnecessary antibiotic use, particularly in high-income countries [12]. Of particular note, the United Nations High-Level Meeting on Antimicrobial Resistance (AMR) convened global leaders to establish international commitments and targets, including a 10% reduction in AMR-related mortality by 2030 and improved antibiotic stewardship in human healthcare [14]. Both interventional and observational studies indicate that AMS initiatives can significantly reduce antibiotic consumption, thereby alleviating the selective pressure that drives resistance. For instance, systematic reviews and meta-analyses suggest that AMS programs can lead to a 15% to 30% reduction in antibiotic use in hospital settings, a decline that is directly associated with lower rates of resistant bacterial strains [14]. AMR is not merely a distant future threat but an urgent global health crisis, with devastating consequences already evident in terms of increased mortality and depleted healthcare resources. Ensuring the rational use of antibiotics remains a cornerstone of global efforts to combat this crisis [15].”
3) Lines 32-35: “some growth” is very arbitrary. I suggest the two sentences to be merged and the authors to focus specifically on examples like the increase in Brazil.
Authors´response: The authors We agree with eliminating vague language and combining sentences for a more direct impact, focusing on concrete examples.
Introduction section: “Additionally, following the Coronavirus Disease 2019 (COVID-19) pandemic, an increase in antibiotic prescriptions was observed in Brazil. A survey conducted by the Federal Pharmacy Council of Brazil indicated that between 2019 and 2020, antibiotic consumption surged by over 60% in the Southeast, Northeast, and Central-West regions, while the North region experienced an alarming 123% rise [9]. Dos Santos et al. [10] further corroborated this trend, reporting a 21.3% increase in antibiotic consumption in 2020 compared to the previous year.”
4) Lines 37-41: The authors have not yet spoken about whether antibiotics are known to have other drug interactions or if there is existing evidence suggesting that there may be. I suggest the authors to include a sentence or two about any existing known drug interactions with antibiotics, (particularly through the mechanism of reducing the resident microbiota) or if there is none known, that the authors identify this and highlight the need for more research.
Authors´response: Thank you for your feedback. We've changed it as suggested.
Introduction section: “While no other medications directly alter gut microbiota through antibiotic interaction, Nonsteroidal Anti-inflammatory Drugs (NSAIDs) and Proton Pump Inhibitors (PPIs) can exacerbate or modulate antibiotic-induced microbiota changes. Administered concurrently or sequentially, these drugs may impact microbial composition, potentially amplifying dysbiosis or affecting gut recovery [22,23].”
5) Line 44 – “whether oral or injectable” – it is not clear whether the authors are referring to antibiotics or hormonal contraceptives here.
Authors´response: We acknowledge ambiguity. The text has been corrected to make clear what it refers to.
Introduction section: “…the concomitant use of penicillin with hormonal contraceptives (whether oral or injectable).”
6) Line 44 – not all hormonal contraceptives are a combination of hormones – for example progesterone-only contraceptives. Please ensure language is clear and accurate throughout.
Authors´response: We appreciate the correction. The language has been adjusted to be more accurate and inclusive regarding types of hormonal contraceptives.
Introduction section: “Hormonal contraceptives can consist of a combination of hormones, such as estrogen and progesterone, or just progesterone. When introduced into the body, these hormones are metabolized in the liver.”
7) Line 54 – did Aronson and Ferner also correct for/ test for confounding effects from the infection itself? Please mention briefly.
Authors´response: We briefly mention Aronson and Ferner's consideration of confounding factors.
Introduction section: “In their study, when analyzing reports of unintended pregnancies, they considered diarrhea as a possible confounding factor in interactions. However, although diarrhea caused by antibiotics might have reduced the effectiveness of the oral contraceptives, it didn't show up as a signal. Therefore, it can't be considered a confounding factor in this data.”
8) Line 57 – are there any reasons why these two studies may have found conflicting results, such as study design? For example, a sentence like “these two studies found different results although it must be noted that study #1 included only X and not Y or Z”.
Authors´response: Thank you for your suggestion. We have added a sentence that briefly discusses the possible reason for the conflicting results, focusing on the study design.
Introduction section: “These two studies [27,28], presented conflicting findings. It is crucial to highlight that this divergence likely stems from fundamental differences in their methodologies. The investigation by Aronson and Ferner [27] was an analysis of reported suspected adverse drug reactions, whereas Simmons et al. [28] conducted a systematic review concentrating on primary studies. Such variations in study design and analytical approaches can account for the observed variability in results.”
9) Line 65: The sentence “The study’s findings intend to shed even greater light…” is redundant, authors can omit.
Authors´response: We agree. The sentence will be removed.
10) Line 66: “The public health crisis” that is referred to – please include references and/or more specific language to back up this statement, as it is very important to the study.
Authors´response: We appreciate the suggestion. References and more precise language regarding the antimicrobial resistance crisis have been incorporated, as this issue is a central focus of the study.
Conclusion section: “Additionally, considering the growing global public health crisis due to antibiotic resistance and polypharmacy, expanding knowledge about the characteristics associated with the prescription of these medications may support the planning of interventions to address the issue [30-32].”
11) Lines 80-84: The authors have here identified other antibiotics in the subclass of beta-lactams, but have not specifically said whether or not these are included in the study – please revise language for clarity.
Authors´response: We appreciate the observation and clarify that the list of penicillin and other beta-lactams presented in the methods section is illustrative. While the review primarily focuses on penicillin, studies addressing other beta-lactams were deemed eligible if their findings aligned with the review’s objectives.
Materials and Methods section: “The use of penicillin was considered a form of exposure. Penicillin belongs to the beta-lactam subclass of antibiotics, which includes, but is not limited to, amoxicillin, ampicillin, carbenicillin, dicloxacillin, nafcillin, oxacillin, penicillin G, penicillin V, piperacillin, and ticarcillin. Studies examining the interaction of hormonal contraceptives with other beta-lactam antibiotics beyond penicillin were also considered, provided their findings were relevant to the broader discussion of interactions within this subclass.
12) Line 137 – I commend the authors use of this flowchart, it is very clear and easy to understand.
Authors´response: We appreciate the compliment. We are pleased that the flowchart was clear and useful.
13) Lines 146 onward – this may be outside of the authors control, but this table needs to be re-formatted as it is extremely difficult to read in its current state (perhaps needs to be made landscape?). Also, first row – “ampiciline” please use English term. I also recommend that a column is included to detail what hormones were included in each contraceptive – eg. Progesterone only, combined pill, etc. This has been included but only for some studies, so please make it consistent throughout the table. May be easier to include a separate column as I have mentioned.
Authors´response: We acknowledge that the tables are difficult to read. We will implement the necessary changes to enhance legibility, including reformatting them to landscape orientation, revising the language, and incorporating the hormone column.
14) Line 148 onward – Row 5 of Table 2 – “the Pill” – this term is not used elsewhere in the article, so please ensure consistent language throughout. Also, the term “OC” is used in this table but not in table 1, please ensure consistency throughout the article for clarity.
Authors´response: We replaced the term "the Pill" with "Drug" in Line 5 of Table 2. In addition, we changed the term oral contraceptive to OC throughout Table 1.
Results section: “…have been taken after an episode which may impair the efficacy of the drug.”
15) Line 153 – It is not clear what is meant by “63 registers”, please clarify.
Authors´response: We clarify what "registers" mean in this context.
Results section: “…from 63 registers of pregnancies in women who received antibiotics…”
16) Line 155 – The sentence “Some data were not possible to be collected in the studies” suggests that the current authors may have had control over what data were collected in the studies. For clarity, I suggest the authors change the wording to something along the lines of “Some of the studies that were evaluated in this review did not provide data of interest, such as…”
Authors´response: Thank you for the suggestion. The sentence will be rephrased for clarity..
Results section: “Some of the studies that were evaluated in this review did not provide data of interest, such as…”
17) Line 161 – Did Bainton show with evidence that circulating estrogen decreases with penicillin? Please clarify in the text, and include how this was done (number of patients, method of measuring hormone levels etc) as this would be important. Did overall circulating sex hormone decrease, or was it just estrogen (note also: the authors also have not addressed the interconversion of estrogen to androgen hormones, which may play into levels of circulating hormone)? Again, please clarify in the text. Furthermore, I suggest the authors avoid use of his/her pronouns and use “their” (i.e. “Bainton [22] associated contraceptive failure in their case report…)
Authors´response: We acknowledge the suggestion for enhanced clarity regarding Bainton's findings and the phenomenon of reduced circulating estrogen. Furthermore, we will rectify the use of pronouns to ensure precision and academic rigor throughout the text.
Results section: “Bainton [36] posited a causal link between contraceptive failure in a reported case and the established mechanism by which penicillin perturbs the gut microbiota, potentially leading to a reduction in circulating estrogen levels, which could subsequently result in escape ovulation. In Bainton's investigation, this association was inferred from a clinical case presentation and an understanding of the proposed mechanism involving the enterohepatic circulation of estrogens. Direct measurement data of hormonal levels were not presented in this specific case report. The intricate interplay between estrogen and androgenic hormones, mediated by enzymes such as aromatase, plays a pivotal role in the regulation of circulating hormone levels. This process significantly influences crucial physiological functions, including sexual development, bone health, and cardiovascular integrity. Dysregulation in this conversion can lead to hormonal imbalances, contributing to conditions such as insulin resistance and osteoporosis. Furthermore, various factors, including pharmacotherapy, endocrine disorders, and alterations in the gut microbiota, can interfere with this enzymatic mechanism, thereby modifying systemic hormone concentrations [44]. While the broader discussion concerning the conversion between estrogens and androgenic hormones was not the primary focus of the studies included in this review, we acknowledge the inherent complexity of hormonal metabolism within this context.”
18) Discussion – I commend that the authors have used this study to highlight a lack of focus into this area, and have consistently shown that although the existing evidence is limited, there must be more work done.
Authors´response: We appreciate the compliment. Our objective is indeed to highlight the research gap and the need for further studies.
19) Lines 198 onward – please clarify what “yellow card reports” are.
Authors´response: We appreciate the feedback. We will clarify the meaning of "yellow card reports.
Discussion section: “Back et al. [43] evaluated 'yellow card' reports—spontaneous notifications of suspected adverse drug reactions submitted by healthcare professionals in the United Kingdom to the Committee on Safety of Medicines between 1968 and 1984. This study may be subject to measurement bias, as these reports are subjective and depend on the perception of the reporting professional.

Reviewer 2 Report
Comments and Suggestions for Authors
The idea of the study is good. It is comprehensively investigated. Here are few suggestions;
- The tables are not appropriately presented. Some of the columns includes too much text and the size is very small which make it difficult to read and interpret for authors.
- The authors have ignored pharmacokinetic studies. Is there any pharmacokinetic study available?
- Some studies have been conducted in animals. They can be included or referred to.
- What are the official recommendations for the co-prescription of antibiotics and contraceptives.
- The conclusion needs to be elaborated.
- Overall, the presentation needs to be improved
- Avoid repetition.
Comments on the Quality of English Language
English can be improved.
Author Response
1) The authors have done a great job at identifying an important area of research that is significantly lacking in data: whether antibiotics impact hormonal contraceptives. Overall, the paper reads well and is very interesting. My comments for improvement mainly focus on use of language and clarity. The tables are not appropriately presented. Some of the columns includes too much text and the size is very small which make it difficult to read and interpret for authors.
Authors’ response: We appreciate the compliment! We agree that the tables require significant improvements in their presentation. The suggestions have been incorporated into the proposed revisions for Reviewer 2, items 13 and 14.
2) The authors have ignored pharmacokinetic studies. Is there any pharmacokinetic study available?
Authors’ response: We recognize the importance of pharmacokinetic studies in understanding interaction mechanisms. Although the scope of this review has focused on studies reporting clinical outcomes, we acknowledge the relevance of mentioning the existence and contribution of these studies. A brief reference to this will be included in the discussion.
Discussion section: “Pharmacokinetic research investigates direct interactions between drugs and metabolites, aiming to elucidate complex metabolic pathways [51]. A comprehensive understanding of these additional mechanisms is essential for refining clinical recommendations and enhancing patient safety. Although pharmacokinetic studies are not the central focus of this scoping review, they remain critical for clarifying the precise pathways by which drugs interact within the human body. Integrating pharmacokinetic findings with clinical outcome data is essential for a more accurate assessment of the risk of drug interactions [52,53].”
3) Some studies have been conducted in animals. They can be included or referred to.
Authors’ response: Our scope was limited to studies involving adolescent, adult, and elderly women, which excluded animal studies from the primary eligibility criteria. However, we may mention the existence of animal studies in the discussion to provide contextualization for the research.
Discussion section: “Additionally, studies employing animal models also contribute significantly to understanding the underlying mechanisms of drug interactions, offering valuable insights into metabolic pathways and the physiological intricacies of the interaction. Nevertheless, these studies were not eligible for direct inclusion in this review due to our exclusive focus on human populations [34].
4) What are the official recommendations for the co-prescription of antibiotics and contraceptives.
Authors’ response: This is an excellent suggestion and a crucial point in terms of clinical relevance. We will incorporate the official recommendations into the discussion.
Discussion section: “Official recommendations for the co-prescription of antibiotics and hormonal contraceptives vary among different health agencies and clinical guidelines. Generally, guidelines recognize that only rifampicin and rifabutin, which are enzyme-inducing antibiotics, have a proven significant impact on the effectiveness of hormonal contraceptives, thus requiring specific counseling or an alternative contraceptive method [57]. For other classes of antibiotics, including penicillin, many health agencies, such as the U.S. Centers for Disease Control and Prevention (CDC) and the UK's Faculty of Sexual and Reproductive Healthcare (FSRH), state that there is insufficient evidence to routinely recommend the use of additional contraceptive methods, although cautious advice to use a barrier method for patient reassurance may be considered [58,59]. Our findings corroborate the inconsistency in the literature regarding penicillin, indicating the need for clearer and more robust evidence-based guidelines.”
5) The conclusion needs to be elaborated.
Authors’ response: We agree that the conclusion could be more detailed and should reinforce the main message of the review.
Conclusion section: “It can thus be concluded that, although the literature suggests a potential drug interaction between hormonal contraceptives and penicillin, the level of scientific evidence is limited and often methodologically weak, with a predominance of case reports and case series. The inconsistency of findings and the prevalence of studies with potential methodological biases restrict the ability to draw definitive conclusions regarding the frequency and clinical impact of this interaction at the population level. In light of these considerations, further research—incorporating systemic and population-level factors and employing more robust study designs (e.g., cohort studies or controlled clinical trials, where ethically feasible)—is both justified and essential to better understand this type of drug interaction and its potential implications, particularly concerning unintended pregnancies, which can have significant consequences in women's lives.”
6) Overall, the presentation needs to be improved and avoid repetition.
Authors’ response: We accept this comment. The suggested improvements regarding table formatting and language clarity, as addressed in the previous responses, aim to enhance the overall presentation of the manuscript.
7) Avoid repetition.
Authors’ response: We appreciate the observation and will carefully review the manuscript to identify and eliminate unnecessary repetitions, ensuring a more concise and fluid reading experience.

Reviewer 3 Report
Comments and Suggestions for Authors
The authors have reviewed the literature on the potential interaction between penicillin and contraceptive methods. This scoping review may provide valuable insights due to the ongoing controversy surrounding this subject.
- The objectives of each study should be mentioned in Table 1, and the table should be restructured in a portrait format for better clarity.
- The duration of the studies should also be included in the table.
- The studies contain information about other medications that may have potential drug interactions, and the authors are encouraged to include this information.
Author Response
1) The authors have reviewed the literature on the potential interaction between penicillin and contraceptive methods. This scoping review may provide valuable insights due to the ongoing controversy surrounding this subject.
Authors’ response: We appreciate the reviewer’s positive assessment. Our primary goal was indeed to map the existing evidence on the potential interaction between penicillin and hormonal contraceptives, given the ongoing controversy and the lack of consensus in clinical practice. We hope that this scoping review contributes to a better understanding of the topic and encourages the development of clearer, evidence-based guidelines.
2) The objectives of each study should be mentioned in Table 1, and the table should be restructured in a portrait format for better clarity.
Authors’ response: We appreciate the suggestion. We will include a dedicated column for study objectives in Table 1 and undertake a comprehensive restructuring to enhance clarity. However, given the substantial amount of information, a landscape format may prove more suitable for Table 1, and we will strive for the optimal possible presentation.
3) The duration of the studies should also be included in the table.
Authors’ response: We accept the suggestion. We will include the study duration in Table 1, where this information is available in the original sources.
4) The studies contain information about other medications that may have potential drug interactions, and the authors are encouraged to include this information.
Authors’ response:
We have include this information on the last paragraph of the Results section.
The analyzed studies presented other medications that may have potential interactions with hormonal contraceptives, but with limited evidence, such as: nitrofurantoin [27], nevirapine [27], rifabutin [27], rifampicin [27], ritonavir [27], citalopram [27], topiramate [27], lansoprazole [27], loperamide [27], loratadine [27], propranolol [27], theophylline [27], zolpidem [27], tetracycline [27, 37, 40, 43], phenytoin [27, 40, 43], phenobarbitone [27, 43], primidone [27, 43], carbamazepine [27, 40, 43], ethosuximide, sodium valproate [43], cotrimoxazole [43], metronidazole [27, 40, 43], cephalosporins [27,43], trimethoprim [27, 43], erythromycin [27, 40, 43], sulphonamides [43], griseofulvin [27, 43], fucidic acid [43], co-trimoxazole [40], isoniazid [40], amphotericin [40], oxazepam [40], azatadine [40], cortisone [40], indomethacin [40], ibuprofen [27, 40, 42], paracetamol [27, 40, 42], aspirin [40], prochlorperazine [40].

Round 2
Reviewer 2 Report
Comments and Suggestions for Authors
The paper is much improved now and publishable if authors pay more attention to formatting and follow the guidelines of the journal. However, the authors should follow standard format for refrenceing. From line 236 to 247, the authors have used multiple time a refrence in a single sentence "furantoin [27], nevirapine [27], rifabutin [27], rifampicin [27], ritonavir [27], citalopram [27......"this refrence can be used once at the end of sentence. Such corrections should be made throughout the manuscript.
In table the authors have mentioned age. It will be better to write age as; Age (years). The serial number 3 and 4, the values are represnted as mean but the data is about only one patient, please remove mean.
Comments on the Quality of English LanguageCan be improved
Author Response
The paper is much improved now and publishable if authors pay more attention to formatting and follow the guidelines of the journal. However, the authors should follow standard format for refrenceing. From line 236 to 247, the authors have used multiple time a refrence in a single sentence "furantoin [27], nevirapine [27], rifabutin [27], rifampicin [27], ritonavir [27], citalopram [27......"this refrence can be used once at the end of sentence. Such corrections should be made throughout the manuscript.
Response: Thanks for your comments. We have corrected the manuscript accordingly.
In table the authors have mentioned age. It will be better to write age as; Age (years). The serial number 3 and 4, the values are represnted as mean but the data is about only one patient, please remove mean.
Response: We have corrected the manuscript accordingly.